# Instruments for assessing back pain in athletes: A systematic review

Vinicius Diniz Azevedo[1]*, Regina Márcia Ferreira Silva[2]*, Silvia Cristina de Carvalho Borges[1], Michele da Silva Valadão Fernades[2], Vicente Miñana-Signes[3], Manuel Monfort-Pañego[3], Priscilla Rayanne E. Silva Noll[2,4], Matias Noll[1,2]

1 Federal University of Goiás, Goiânia, Brazil, 2 Goiano Federal Institute, Itumbiara, Brazil, 3 University of Valencia, Valencia, Spain, 4 University of São Paulo, São Paulo, Brazil

* regina.silva@ifg.edu.br (RMFS); vinicius.diniz.azevedo@hotmail.com (VDA)

**Data Availability Statement:** All relevant data are within the paper and its Supporting Information files.

## Abstract

Back pain in athletes varies with sport, age, and sex, which can impair athletic performance, thereby contributing to retirement. Studies on back pain in this population use questionnaires to assess components, such as pain intensity and location and factors associated with pain, among others. This study aimed to review validated questionnaires that have assessed back pain in athletes. This systematic review was conducted according to Preferred Reporting Items for Systematic Reviews and Meta-Analyses (PRISMA) by searching the databases Embase, MEDLINE, SPORTDiscus, CINAHL, and Scopus. The articles were selected regardless of language and date of publication. Titles and abstracts were independently selected by two reviewers; disagreements were resolved by a third reviewer. All the steps were conducted using the software Rayyan. The methodological quality of the questionnaire validation articles was assessed using a critical appraisal tool checklist proposed by Brink and Louw. The search returned 4748 articles, of which 60 were selected for this review, including 5 questionnaire validation studies. These articles were published between 2004 and 2022, which were performed in more than 20 countries, particularly Germany (14) and Sweden (5). Thirteen different instruments were identified, of which 46.1% were developed in Europe. The most commonly used questionnaires were the Oswestry Disability Index and Nordic Standardized Questionnaire. In addition, five questionnaire validation studies were selected for methodological quality assessment, with only two studies demonstrating high methodological quality. The following three instruments were identified for assessing back pain specifically in athletes: Micheli Functional Scale, Persian Functional Rating Index, and Athlete Disability Index. This review confirmed that all three instruments were specifically designed to assess this condition.

## 1 Introduction

Pain is defined by the International Association for the Study of Pain (IASP) as "an unpleasant sensory and emotional experience associated with, or resembling that associated with, actual or potential tissue damage [1]." Musculoskeletal pain has been widely researched in the

**Funding:** The author(s) received no specific funding for this work.

**Competing interests:** The authors have declared that no competing interests exist.

scientific literature [2] and may occur in various areas of the body, such as the back. Back pain [(BP)] can be conceptualized as, "existing or non-existing, present or previous pain of any kind in the thoraco-lumbar spine [3]" or even as "pain in the cervical, thoracic and/or lumbar areas [4–7]." Thus, controlling BP is highly relevant and crucial, particularly for athletes [8] Soon, sustained pain results in a decline in athletic performance, which makes it difficult to maintain a competitive level of play [9].

BP in athletes is associated with different factors, such as sleep [10–12], high training volumes [13], sex [14, 15], sport [16, 17], and psychosocial dimensions [18], and other factors [19]. This condition has been studied in sports, such as rowing, cycling, gymnastics, shooting, rugby, and football [20–23]. Therefore, medical literature has increasingly recognized the need to consider BP a disease, and therefore, evaluate it [24]. Pain assessment is so important that if we ask the individual if he has pain, his simple answer is not enough for reliability of response [25].

Most (observational and experimental) epidemiological studies aimed at assessing BP and associated risk factors are based on questionnaires [26, 27]. For example, the Visual Analog Scale (VAS), the Numerical Rating Scale (NRS), and the Pain Severity subscale of the Brief Pain Inventory (BPI-PS) are the most commonly used instruments to measure pain intensity in low back pain [28] and the first two are considered reliable in assessing low back pain severity [29]. Even assessing pain intensity using verbal rating scales and visual analog scales has been considered valid for many years [30]. Operationally, these methods are used to evaluate a research object and measure the characteristics and/or attributes related to the phenomena, individual, process, or organization [31]. Developing a questionnaire is a rigorous and complex process, involving writing, ordering, and presenting items [32].

Many questionnaires assessing BP are prevalent in different populations [2, 33, 34]. However, athletes are subject to more intense and sport-specific variables, which are not included in these questionnaires [35]. Therefore, certain questionnaires have limitations; however, efforts have been made to develop specific questionnaires for athletes [10, 22, 36–39]. However, despite these efforts, no instrument is considered the gold standard for athletes [13].

This study is relevant because the demand in health care, combined with the need to use evaluative instruments, has contributed to increase, for example, the pressure on health care professionals to ensure the implementation of evidence-based practice. Thus, publication of systematic review studies, as well as others that synthesize research results, is an important action for evidence-based practice. Moreover, this type of study serves to guide the development of projects, indicating new directions for future investigations. Therefore, the aim of this study was to review the scientific literature related to validated questionnaires to evaluate BP in athletes. The hypothesis is that there are few valid and reliable instruments to evaluate BP in athletes and that the existing ones contribute little to the specific assessment.

## 2 Materials and methods

### 2.1 Protocol and registration

To support this review, it was performed and published an article on a systematic review protocol entitled: "*Evaluating Instruments for Assessing Back Pain in Athletes: A Systematic Review Protocol*" [35].

This systematic review was conducted according to Preferred Reporting Items for Systematic Reviews and Meta-Analyses (PRISMA) [40]. To increase transparency and reproducibility, while avoiding duplication of efforts on the same topic, this review was submitted to and registered in the International Register of Prospective Systematic Reviews (PROSPERO) under CRD42020201299. Ethical approval was not required because this study did not involve human participants.

## 2.2 Research question identification

The main question of the study was "what are the existing, valid, or reliable instruments for assessing BP in athletes?"

## 2.3 Search strategy and eligibility criteria

The initial search for articles was performed by one researcher (VDA) on October 27, 2022, using the following five databases: MEDLINE, Scopus, Embase, SPORTDiscus, and CINAHL. Search terms were combined using Boolean operators (AND/OR), and searches for articles were performed in the databases without restricting the year, country, and language of publication. Box 1 presents the logical structure of the general search strategy with the Boolean descriptors and operators used in the databases. The search strategy for each database is presented in S1 Annex.

### Box 1. Electronic search strategy

| |
|---|
| (#1) *"Pain measurement"* OR *"Questionnaire"* OR *"instrument"* OR *"form"* OR *"assessment"* OR *"score"* OR *"measurement"* OR *"scale"* OR *"tool"* |
| (#2) *"Back pain"* OR *"low back pain"* OR *lumbago* OR *"neck pain"* OR *backache* OR *"spinal pain"* OR *"neck ache"* OR *"neck pain"* |
| (#3) *"Athlete"* OR *"sport"* OR *"sportsman"* OR *"sportswoman"* |
| (#1) AND (#2) AND (#3) |

This review included articles whose instruments met the following criteria: a) mentioned prevalence, incidence, intensity, location, functional disability, or other BP-related components, b) assessed athletes and sports-related variables, and c) could be created, adapted, or translated but were validated or at least tested for their reliability. In this research, validated questionnaires were analyzed regardless of the country. Therefore, the same questionnaire could be validated for athletes in different countries and may thus vary based on cultural adaptation.

Thus, the articles that met the following criteria were excluded from this review: a) systematic reviews, reports, opinion articles, response letters, and book chapters, b) research that included individuals with physical or mental disabilities, pregnant or lactating women, and participants with spinal fractures or who have recently undergone surgery, c) studies conducted in specific and/or traditional communities (for example, rural communities, indigenous populations, refugees, uncontacted individuals, and remote and isolated communities).

The methodological quality was assessed by selecting articles that evaluated the measurement properties of instruments specifically developed for athletes. In this review, the term BP was defined as "cervical, thoracic and/or lumbar pain [4–7]," and athletes were defined as "individuals who participate in athletic competitions and are involved in training activities for four or more hours per week [41–43]."

## 2.4 Study review and selection process

The results from the search strategies were imported into the Mendeley software; duplicate articles were identified and removed. The first selection phase consisted of reading the title and abstracts to assess whether each article met the eligibility criteria. After this phase, the

articles were read completely to confirm their eligibility. All the steps were performed using the Rayyan software, which is specifically designed for conducting systematic reviews [44].

The article-selection step was independently performed by two reviewers, and disagreements were resolved by a third reviewer. Interrater reliability for individual component ratings was determined by calculating the percent agreement and Cohen's Kappa coefficient. Thereafter, eligible articles were included in this systematic review.

### 2.5 Methodological quality

The methodological quality of the articles that evaluated the measurement properties of the instruments specifically developed for athletes was determined using the Critical Appraisal Tool (CAT) proposed by Brink and Louw [45]. The scale consists of 13 items, of which five refer to both validity and reliability, four to validity, and four to reliability studies. Each item is scored as "Yes," "No," or "Not Applicable (N/A)." This scale was used by the same independent reviewers. A study is considered of high methodological quality when the score $\geq 60\%$ [46, 47].

### 2.6 Reviewer training

Before starting the search process, the reviewers who participated in the eligibility assessments were trained concerning the inclusion/exclusion criteria of the study, which included a practical assessment of the eligibility of 50 abstracts [7]. Lastly, the training process covered aspects such as correctly using the software Rayyan and standardizing procedures. The reviewers were trained on how to use the software to use the same selection criteria for the articles. In this case, at first, a reviewer included in the software a file with data from the articles and began the selection of those that fit the selection criteria by reading the titles and abstracts, including justifying the exclusions. Later, the second reviewer performed the same procedure.

### 2.7 Data extraction

The following data were extracted from the selected articles: author and year of publication, country of research, participants' age, name of the evaluation instruments, sport, sample size, level of physical activity, and BP definition. The following data were extracted from the selected instruments: type, name, number of items of the instrument, assessment method and objective of the assessment instrument. The following data were extracted from the studies selected for methodological quality assessment: study (author and year), instrument objective, name (instrument abbreviation), sport/activities assessed by the instrument, validity, internal consistency and reliability.

## 3 Results

The PRISMA selection process and flow diagram of this systematic review is shown in Fig 1. In total, 4,748 articles were imported from five databases (Embase, MEDLINE, SPORTDiscus, CINAHL, and Scopus). Of this total, 2,136 duplicates were removed, and 2,376 articles were excluded after reading the title and abstract. The remaining 236 articles were read completely, subsequently excluding 176 articles. Ultimately, 60 articles were selected for this review. Upon article selection, the inter-rater reliability was 96%, with a Cohen's kappa of 0.53.

In total, 60 articles were selected for this review, including 5 athlete questionnaire validation studies. The articles were published between 2004 and 2022, and the sample size ranged from 8 [48] to 2,116 [49] participants. The participants' age ranged from 10-year-old gymnasts [50] to 56-year-old golfers [51]. The terms that defined the athletes' competitive level, such as

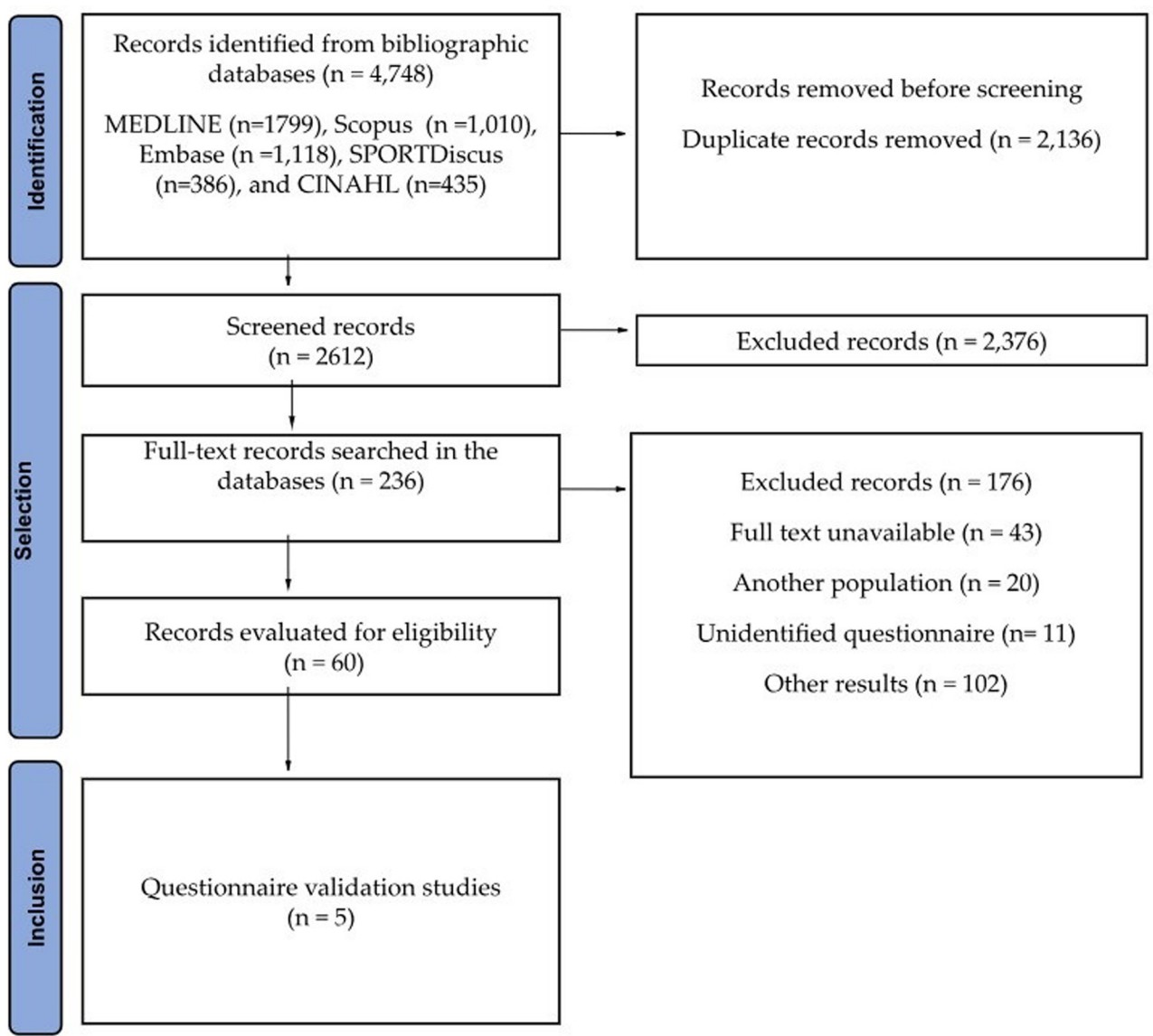

**Fig 1. Preferred Reporting Items for Systematic Reviews and Meta-Analyses (PRISMA) flow diagram of articles included in this review.**

educational (education, university student, elite sports school), divisional (international, national, regional, and local), age (veterans), competition (professional or elite and recreational (club and intramural sports and Junior Olympic program) levels varied considerably between the studies.

Of the 60 articles, most were surveyed in athletes in Germany [14, 18, 49, 52–62], in Sweden [3, 63–66], and in Iran [36, 37, 67–69]. Moreover, also in Norway [11, 70, 71], Poland [72–74], United States [50, 51, 75], the Netherlands [76, 77], Belgium [78], Switzerland [12, 79], Finland [80, 81], Australia [82, 83], Spain [84, 85], Saudi Arabia [86, 87], Estonia [88], Serbia [89], New Zealand [90], Ukraine [91], China [92], Austria [93], Brazil [94], and India [95]. Three articles [48, 96, 97] failed to report the study site, and one study was conducted in several countries [98]. In addition, only 23 studies defined BP, whereas 37 studies did not. Further details are provided in Tables 1 and 2.

**Table 1. Summary of the article included in this review.**

| Reference | Country | Age, mean, standard deviation | Sample | Instrument | Sport | Level of competition | Backpain definition |
|---|---|---|---|---|---|---|---|
| Bahr, Andersen et al. 2004 [70] | Norway | 23 ± 5 (M) 21 ± 4 (F) 21 ± 6 (M) 22 ± 5 (F) 24 ± 7 (M) 23 ± 6 (F) | 781 | SNQ | Skiing, Rowing and Orienteering | Elite | Yes |
| Hanrahan, Van Lunen et al. 2005 [96] | – | 20.3 | 19 | MPQ | Athletics | Collegiate | No |
| Kraft, Scharfstädt et al. 2007 [52] | Germany | 21.9 | 20 | ODI | Equestrian vaulting | Elite | No |
| Baranto, Hellström et al. 2009 [63] | Sweden | 26 (athletes) and 28 (non-athletes) 40 (athletes) and 42 (non-athletes) | 92 | ODI | Several sports | Elite | Yes |
| Tsai, Sell et al. 2010 [51] | United States | 47.9±8.3 with pain 48.6 ±7.4 without pain | 32 | Modified ODI | Golf | – | No |
| Clarsen, Krosshaug et al. 2010 [98] | Several | 26 | 109 | SNQ | Cycling | Professional | Yes |
| Jonasson, Halldin et al. 2011 [79] | Switzerland | 28 (athletes) 21 (non-athletes) | 77 | ODI | Several sports | Elite | No |
| John Kachanathu, Zakaria et al. 2012 [87] | Saudi Arabia | 20.7 ± 2.0 | 30 | ODI | Cricket -Fast bowlers | Professional | No |
| Hilgersom, Kuilman et al. 2012 [76] | The Netherlands | 18 ± 14–25 | 75 | ALBPSQ-DLV, RMDQ, ODI, CPG and SNQ | Speed skating | Elite | Yes |
| Sutton, Guin et al. 2012 [48] | – | 19–21 | 8 | ODI | Several sports | Collegiate | No |
| Cole and Grimshaw 2014 [82] | Australia | 46 (with pain) 39 (without pain) | 27 | SF- MPQ | Golf | – | No |
| kuotras, Buecking et al. 2014 [53] | Germany | 42±15 | 137 | SNQ | Auto racing | Professional | No |
| Corkery, O'Rourke et al. 2014 [75] | United States | 21.2 ± 2 (with pain) 20.87 ±1.3 (Control group) | 15 | ODI | Several sports | – | No |
| Külling, Florianz et al. 2014 [93] | Austria | 28 ± 19–39 | 29 | RMDQ and ODI | Beach volleyball | Elite | No |
| Ng, Cañeiro et al. 2015 [83] | Australia | 15.2 ± 1.5 (Control)] 16.3 ± 1.5 (Intervention) | 36 | RMDQ | Rowing | School and Community Clubs | No |
| Tunås, Nilstad et al. 2015 [11] | Norway | 22.4 ± 4 soccer 22.3 ± 3 handball | 467 | SNQ | Soccer and Handball | Elite | Yes |
| van Hilst, Hilgersom et al. 2015 [77] | The Netherlands | 16 ± 14–19 field hokey (F) 17± 15–24 field hokey (M) 18 ± 14–25 Speed skating (F) 18 ± 15–23 Speed skating (M) 18 ± 16–19 Soccer (M) | 181 | SNQ and ALBPSQ-DLV | Field Hockey Speed skating Soccer | Elite | Yes |
| Heidari, Mierswa et al. 2016 [54] | Germany | 28.7 ± 9.7 (athletes) 41.6 ± 14.4 (non-athletes) | 264 | CPGS | Several sports | International National Regional Local | No |
| Heidari, Mierswa et al. 2016 [55] | Germany | 32.24 ± 11.32 | 139 | CPG | Several sports | Competitive Recreational | No |
| Mueller, Mueller et al. 2016 [56] | Germany | 13.2 ± 1.4 | 321 | FPS | Several sports | Collegiate | Yes |

*(Continued)*

**Table 1.** (Continued)

| Reference | Country | Age, mean, standard deviation | Sample | Instrument | Sport | Level of competition | Backpain definition |
|---|---|---|---|---|---|---|---|
| Alricsson, Björklund et al. 2016 [64] | Sweden | 16–19 | 51 | CPG | Skiing | International and national | Yes |
| Pasanen, Rossi et al. 2016 [80] | Finland | 15.8± 1.9 | 401 | SNQ | Floorball and Basketball | – | Yes |
| Abdelraouf and Abdel-aziem 2016 [86] | Saudi Arabia | 21.5 ± 2.5 athletes with pain 22.6 ± 2.42 athletes without pain | 55 | MFS | Several sports | Collegiate | No |
| Clay, Mansell et al. 2016 [97] | – | 19.2 ± 1.1 (Technical sports) 19.5±1.2 (Endurance athletes) | 37 | ODI | Rowing | Collegiate | No |
| Bussey, Kennedy et al. 2016 [90] | New Zealand | 19.3 ± 1.4 with pain 20 ± 1.6 without pain | 39 | ODI | Field hockey | Elite | No |
| Mueller, Mueller et al. 2017 [57] | Germany | 13.2 ± 1.6 | 1559 | FPS | Several sports | Elite sports schools | No |
| Müller, Müller et al. 2017 [49] | Germany | 13.3 ± 1.7 | 2116 | FPS | Several sports | Elite sports schools | Yes |
| Wippert, Puschmann et al. 2017 [18] | Germany | 39± 13 | 588 | CPG | – | Recreational athletes and non-athletes | No |
| Fett, Trompeter et al. 2017 [14] | Germany | 20.9 ± 4.8 | 1.114 | CPG and SNQ | Several sports | Elite | Yes |
| Kums, Ereline et al. 2017 [88] | Estonia | 14.3 ± 1.3 (with back pain) 14.6±1.5 (without back pain) | 32 | ODI | Rhythmic gymnastics | Elite | No |
| Noll, Silveira et al. 2017 [94] | Brazil | 14–20 | 251 | BACKPEI | Volleyball, Basketball, Handball and Soccer | High school athletes | No |
| Thoreson, Kovac et al. 2017 [65] | Sweden | 17.6 ± 1.02 (Skiers]) 16.4 ±0.57 (Control) | 44 | ODI | Skiing | Elite | Yes |
| Rossi, Pasanen et al. 2018 [81] | Finland | 14.9 ± 1.6 (Floorball) 16.8 ± 1.6 (Basketball) | 396 | SNQ | Floorball and basketball | – | Yes |
| Trompeter, Fett et al. 2018 [58] | Germany | 20.9 ± 4.8 | 1.114 | CPG and SNQ | Several sports | Elite | No |
| Todd, Aminoff et al. 2018 [66] | Sweden | 18.2± 1.1 (Skiers]) 16.4 ± 0.6 (Control) | 102 | ODI | Skiing | Elite | No |
| Witwit, Kovac et al. 2018 [3] | Sweden | 18.2 ± 1.1 (Skiers) 16.4±0.6 (Control) | 75 | ODI | Skiing | Elite | Yes |
| Gajsar, Titze et al. 2019 [59] | Germany | 28.6 ± 9.69 (Athletes) 41.6± 14.3 (Non-athletes) | 266 | CPGS | Several sports | International and Regional | Yes |
| Fett, Trompeter et al. 2019 [62] | Germany | 19.7 ± 4.7 (Athletes) 21.2 ± 2.0 (Control) | 347 | CPG and SNQ | Several sports | Elite | Yes |
| Trompeter, Fett et al. 2019 [60] | Germany | 20.7 ± 3.8 (Elite) 25.3 ± 6.1 (Non-elite) | 322 | CPG and SNQ | Rowing | Elite | Yes |
| Sweeney, Potter et al. 2019 [50] | United States | 13.3 ± 2.5 (with pain) 13.8 ± 3.0 (without pain) | 29 | MFS and ODI | Gymnastics | Junior Olympic Program | No |
| Lukas and German 2019 [12] | Switzerland | 19.6 ± 3.5 (Elite technical cycling disciplines) 19.5 ± 5.8 (Elite endurance cycling disciplines) | 111 | ODI | Cycling | Elite | No |

(*Continued*)

**Table 1.** (Continued)

| Reference | Country | Age, mean, standard deviation | Sample | Instrument | Sport | Level of competition | Backpain definition |
|---|---|---|---|---|---|---|---|
| Honcharov, Ruban et al. 2020 [91] | Ukraine | 36–45 | 34 | ODI | Wrestling | Veteran athletes | No |
| Madić, Obradović et al. 2020 [89] | Serbia | 20.4 ± 3.7 | 136 | RMDQ | Soccer | Semi-professional | No |
| Levenig, Kellmann et al. 2020 [71] | Norway | 28.69 ± 9.60 39.34 ± 12.63 | 238 | CPG | - | Elite | Yes |
| Cejudo, Moreno-Alcaraz et al. 2020 [84] | Spain | 22.50 ± 2.89 | 26 | SNQ | Inline hockey | Professional | Yes |
| Błach, Klimek et al. 2020 [72] | Poland | 18–44 | 100 | ODI | Karate | Elite | No |
| Zhou and Fu 2021 [92] | China | 40–50 | 106 | RMDQ | Golf | - | Yes |
| Deckers, De Bruyne et al. 2021 [78] | Belgium | 25 ± 7 | 32 | ODI | Equestrianism | Professional | Yes |
| Sharma, Akmal et al. 2021 [61] | Germany | 21.22 ± 3.41 24.37 ± 3.02 24.38 ± 3.61 | 35 | ODI | Cricket, hockey, volleyball and basketball | University athletes | Yes |
| Zaworski, Gawlik et al. 2021 [73] | Poland | 19–25 21.7 ± 1.8 | 18 | VAS , ODI, BPFS (Back Pain Functional Scale), Modified Laitinen Pain Questionnaire | Soccer | University athletes | No |
| Deckers, De Bruyne et al. 2021 [78] | Belgium | 25 ±7 | 32 | ODI | Equestrianism | National and regional | Yes |
| Sędek, Truszczyńska-Baszak et al. 2022 [74] | Poland | 31.7 ± 5.2 (study) 29.7 ± 4.4 (control) | 61 | ODI | Brazilian Jiu-Jitsu | - | No |
| Kazemkhani, ShahAli et al. 2022 [69] | Iran | 18.87± 1.95 (control) 19.81± 3.7 (study) | 56 | ADI | Several sports | - | No |
| Marugán-Rubio, Chicharro et al. 2022) [85] | Spain | 33.15 ± 7.79 | 80 | RMDQ | Soccer | Semi-professional | No |
| Pavana, K et al. 2022 [95] | India | 18–25 | 50 | SNQ | Volleyball | Collegiate | No |

M: Male, F: Female, ALBPSQ-DLV: Acute Low Back Pain Screenings Questionnaire-Dutch Language Version, BACKPEI: Back Pain and Body Posture Evaluation Instrument, CPG: Chronic Pain Grade Questionnaire, CPGS: Chronic Pain Grade Scale, FPS: Faces Pain Scale, MFS: Micheli Functional Scale, MPQ: McGill Pain Questionnaire, ODI: Oswestry Disability Index, RMDQ: Roland–Morris Disability Questionnaire, SF-MPQ: McGill Pain Questionnaire Short Form, SNQ: Nordic Standardized Questionnaire.

Of the total number of articles included in this review, only 5 were validated questionnaires. These questionnaires were developed and validated for athletes; therefore, their articles were selected for methodological quality assessment. Each of these 5 articles validated a single questionnaire, including the Micheli Functional Scale (MFS) [36, 68], Persian Functional Rating Index (PFRI) [37, 67], and Athlete Disability Index (ADI) [22].

All the five studies assessed validity and reliability. Two studies [36] did not report in which sports the tests were performed. For criterion validity, MFS demonstrated excellent correlation with the Oswestry gold standard Low BP Disability Questionnaire (ODQ) and significantly identified BP in younger and older patients and among female and male patients, thereby indicating that MFS is a valid and widely applicable tool. The total score of the Persian version of

**Table 2. Summary of the instrument validation articles and their questionnaires.**

| Study | Objective | Instrument | Sport (%) | Item | Validity | Corrected Instrument | Internal Consistency | Reliability |
|---|---|---|---|---|---|---|---|---|
| | | | | | | Questionnaire summary | | |
| d'Hemecourt, Zurakowski et al. 2012 [36] | To validate a specific instrument for the functional evaluation of low back pain in the young athlete. | MFS | NR | 5 | r = 0.90 | ODQ | Cronbach's α = 0.904 | Cronbach's α = 0.904 |
| Naghdi, Nakhostin Ansari et al. 2015 [37] | To determine the reliability and validity of an instrument for assessing low-back pain in athletes. | PFRI | Bodybuilding (38%), Aerobics (13.3%), Swimming (12%), Soccer (8%), Judo (8%), Badminton (7.3%), Pilates (6.7%), Others (6.3%). | 10 | r = 0.72 r = 0.83 | NRS PRMDQ | Cronbach's α = 0.90 | Test-retest ICC = 0.97 |
| Naghdi, Ansari et al. 2016 [67] | To validate an instrument for assessing neck pain in athletes. | PFRI | Bodybuilding (30.7%), Aerobics (18%), Swimming (10.7%), Karate (11.3%), Taekwondo (8.7%), Volleyball (6.7%), Soccer (6%), Yoga (4%) Badminton (4%). | 10 | r = 0.94 r = 0.995 | NRS NDI | Cronbach's α = 0.97 | Test-retest ICC = 0.96 |
| Noormohammadpour, Khezri et al. 2018 [22] | To evaluate the validity and reliability of a new proposed questionnaire for assessing functional disability in athletes with low-back pain. | ADI | NR | 12 | r = 0.626 r = 0.918 r = 0.669 | VAS ODI RMDQ | Cronbach's α = 0.91 | Test-retest ICC = 0.74–0.95 |
| Naghdi, Ansari et al. 2015 [68] | To translate and cross-culturally adapt MFS. | MFS | Physical Fitness (28.7%), soccer (24.7%), volleyball (13.3%), judo (9.3%), basketball (8%) and others (16%). | 5 | r = 0.82 r = 0.92 | PFRI VAS | Cronbach's α = 0.73 | ICC = 0.99 |

ADI: Athlete Disability Index, FRI: Functional Rating Index, ICC: Intraclass Correlation Coefficient, MFS: Micheli Functional Scale, NDI: Neck Disability Index, NR: Not Reported, NRS: Numerical Rating Scale, ODI: Oswestry Disability Index, ODQ: Oswestry Low Back Pain Disability Questionnaire, PFRI: Persian Functional Rating Index, PRMDQ: Persian Roland–Morris Disability Questionnaire, r: Pearson product-moment correlation coefficient, rho: Spearman's rank correlation coefficient expressing convergent validity, RMDQ: Roland–Morris Disability Questionnaire, VAS: Visual Analog Scale.

this instrument also significantly correlated with the total score of the visual analog scale (VAS) and PFRI.

The total score of PFRI positively correlated with the total score of the Persian Roland–Morris Disability Questionnaire (PRMDQ) [99]. These data indicate the construct validity of PFRI in athletes with low BP [37]. PFRI also strongly correlated with the Numerical Rating Scale (NRS) and the Persian Neck Disability Index (NDI) [67], indicating the construct validity of the PFRI in athletes with neck pain.

Moreover, the ADI significantly correlated with both the Oswestry Disability Index (ODI) and the Roland–Morris Disability Questionnaire (RMDQ) [22]; therefore, all 3 questionnaires identified athletes with a low BP disability and rated the levels of severity of their disabilities. Moreover, only MFS was identified as an instrument used in athletes in our review. All the 3 questionnaires had high reliability and validity scores, as shown in Table 2.

This review identified 13 questionnaires. The ODI [100] and Nordic Standardized Questionnaire (SNQ) [101] were the two instruments most frequently used in the studies. This reseacrh identified 24 (43.7%) articles using ODI and 14 (25.4%) using SNQ. ODI is an index derived from the instrument Oswestry Low BP Disability Questionnaire (OLBPDQ) [100], developed by Fairbank and Pynsent. A high rate of usability of these instruments was also

reported by another systematic review [13]. The Short-Form McGill Pain Questionnaire (SF-MPQ) [102] is a shorter version of the original MPQ [103]; both questionnaires were developed in Canada and by the same authors. Information on the questionnaires identified in this review is presented in Box 2.

## Box 2. Instruments found in this review

| Assessment instrument | Objective | Assessment | Items of the instruments | Articles that used the instrument |
|---|---|---|---|---|
| McGill Pain Questionnaire Short Form–SF-MPQ [102] | To briefly and quickly assess an individual with significant pain | Questionnaire with 2 subscales: a sensory subscale with 11 items and an affective subscale with 4 items. Each item is rated on a 4-point intensity scale ranging from 0 (none) to 3 (severe pain). | 15 | [82] |
| Chronic Pain Grade Questionnaire–CPG [104] | To grade pain and pain-related disability | Scale with 3 items assessing pain intensity, 3 items assessing pain-related disability and 1 item assessing pain persistence expressed as days in pain | 7 | [14, 18, 55, 58, 60, 62, 64, 71, 76] |
| Faces Pain Scale–FPS [105] | To grade pain | Scale with 5 faces, wherein face 1 corresponds to absence of pain, and face 5 corresponds to maximum pain. | 5 | [49, 56, 57] |
| Chronic Pain Rating Scale–CPGS [106] | To assess pain intensity and disability | Scale with 3 items assessing pain intensity and 4 items assessing disability | 7 | [54, 59] |
| Micheli Functional Scale–MFS [36] | To assess symptoms of low-back pain and ease or difficulty in performing various sporting activities | Questionnaire with 5 items, namely 1 item assessing symptoms, 3 items assessing sporting activities, namely extension, flexion, and jumping, and one item with a visual analog scale scored on 10 points based off a 10-cm line. | 5 | [50, 86] |
| Roland–Morris Disability questionnaire–RMDQ [107] | To assess self-rated physical disability caused by low back pain | Intensity scores are summed and can range from zero (no disability) to 24 (severe disability). Scores higher than 14 points indicate physical disability. | 24 | [76, 83, 85, 89, 92, 93] |
| Acute Low Back pain Screening Questionnaire-Dutch Language Version–ALBPSQ-DLV [108] | To screen for psychosocial risk factors of chronic low-back pain | Questionnaire with 21 items and encompassing the following 5 domains: pain, function, psychological, fear avoidance beliefs, and miscellaneous | 21 | [76, 77] |
| Back Pain and Body Posture Evaluation Instrument–BackPEI [109] | To assess body posture, physical activity level, and prevalence of pain, among other factors | Questionnaire with 20 items expressing the level of exposure to risk factors | 21 | [94] |
| Nordic Standardized Questionnaire–SNQ [101] | To identify musculoskeletal disorders in an ergonomic context | Questionnaire consisting of two sections, namely a general section, which assesses pain or discomfort in nine anatomical regions, and a specific section, which assesses the severity of symptoms in the last 12 months | 17 | [11, 14, 53, 58, 60, 62, 70, 76, 77, 80, 81, 84, 95, 98] |
| Oswestry Disability Index–ODI [100] | To identify musculoskeletal disorders | Questionnaire examining the level of disability in 10 activities of daily living. Each item consists of six statements which are scored from 0 to 5. | 10 | [3, 12, 48, 50–52, 61, 63, 65, 66, 72–76, 78, 79, 87, 88, 90, 91, 93, 97] |
| McGill Pain Questionnaire MPQ [103] | To evaluate a person with significant pain | Questionnaire with 4 groups, 20 subgroups and 78 descriptors, whereby the pain index is calculated as the sum of the intensity values of the applicable descriptors | 78 | [96] |
| Athletes Disability Index–ADI [22] | To assess back pain in sports | Questionnaire with 12 items assessing disability in activities of daily living, such as stretching, strengthening/ weight training exercises | 12 | [69] |
| Back Pain Functional Scale [110] | To assess pain intensity | Contains 12 questions about activities of daily living | 12 | [73] |

ADI: Athlete Disability Index, ALBPSQ-DLV: Acute Low Back Pain Screenings Questionnaire-Dutch Language Version, BACKPEI: Back Pain And Body Posture Evaluation Instrument, BPFS: Back Pain Functional Scale, CPG: Chronic Pain Grade Questionnaire, CPGS: Chronic Pain Grade Scale, FPS: Faces Pain Scale, MFS: Micheli Functional Scale, MPQ: McGill Pain Questionnaire, ODI: Oswestry Disability Index, RMDQ: Roland–Morris Disability Questionnaire, SF-MPQ: McGill Pain Questionnaire Short Form, SNQ: Nordic Standardized Questionnaire.

Six instruments were developed in Europe (RMDQ, ODI, SNQ, Faces Pain Scale, Chronic Pain Grade Scale, Acute Low BP Screenings Questionnaire- Dutch Language Version), five in North America (MFS, CPG, MPQ, SF-MPQ, BP Functional Scale), and only one in South America (BP and Body Posture Evaluation Instrument) and in Asia (ADI). The number of items of the instruments ranged from 5 [36, 105] to 74 [103]. Fig 2 illustrates the origin of each instrument.

### 3.1 Methodological quality

The methodological quality of 2 articles was rated as high (score > 60%) (Table 3). The mean score of the methodological quality assessment was 55.5%. The inter- (item 4) and intra- (item 5) rater blindness, randomization (item 6), and period between repeated measures (item 8) of the items were not assessed because they are not applicable to validity studies.

The items clarifying the rater's qualifications or competence (item 2) and describing cases of sample loss (item 12) were listed in all the articles as "no," i.e., classified as methodological shortcomings. These results prove that, despite the high rate of agreement with other instruments, all the articles failed to report or clarify key information concerning the instrument development process (validity and reliability) and, therefore, lack relevant methodological data for this quality assessment process. Considering the other items, number 1 is related to the

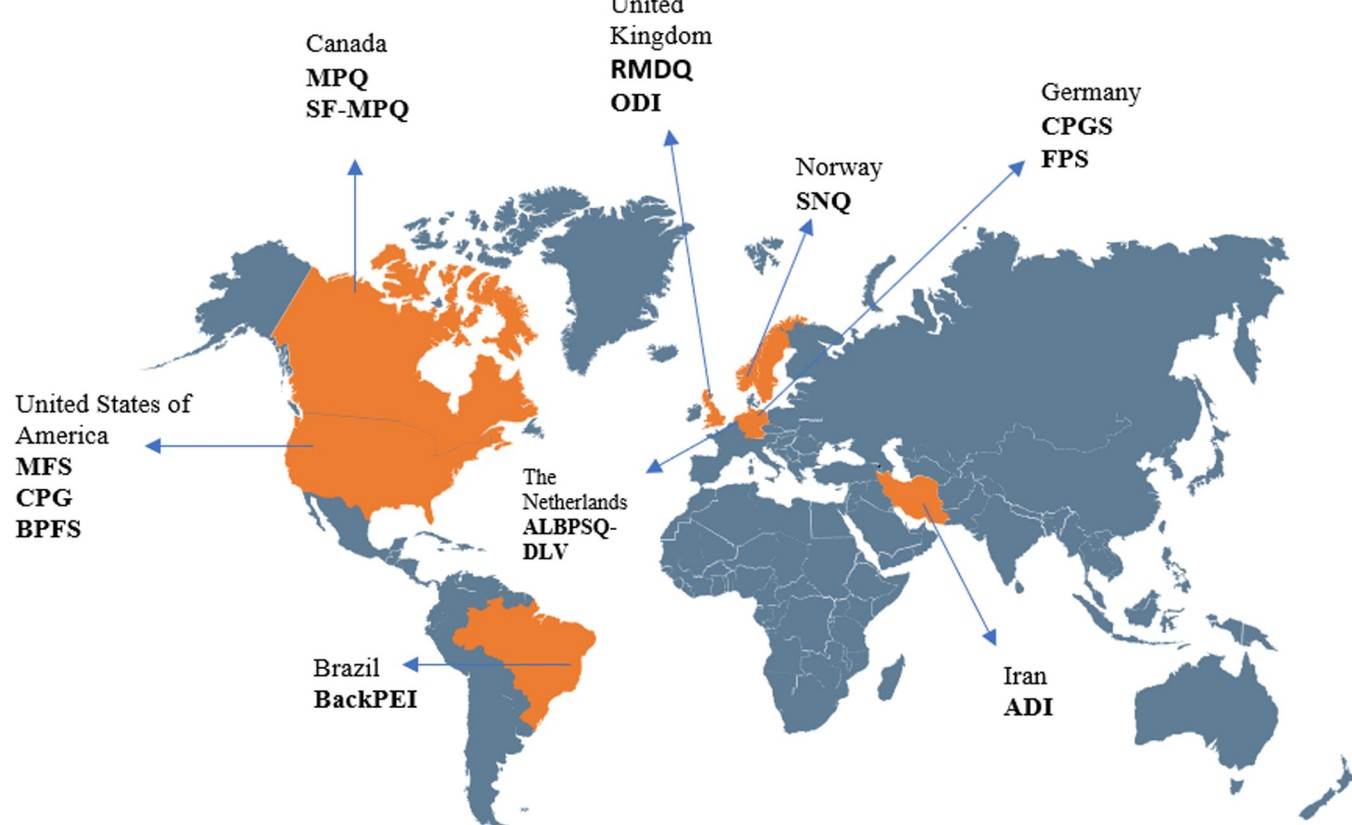

**Fig 2. Origin of each instrument.** ADI: Athlete Disability Index, ALBPSQ-DLV: Acute Low Back Pain Screenings Questionnaire-Dutch Language Version, BACKPEI: Back Pain And Body Posture Evaluation Instrument, BPFS: Back Pain Functional Scale, CPG: Chronic Pain Grade Questionnaire, CPGS: Chronic Pain Rating Scale, MFS: Micheli Functional Scale, MPQ: McGill Pain Questionnaire, ODI: Oswestry Disability Index, RMDQ: Roland–Morris Disability Questionnaire, SF-MPQ: McGill Pain Questionnaire Short Form, SNQ: Nordic Standardized Questionnaire.

**Table 3. Results from the methodological quality assessment of the articles selected in this review.**

| Study (author and year of publication) | 1 | 2 | 3 | 4 | 5 | 6 | 7 | 8 | 9 | 10 | 11 | 12 | 13 | % |
|---|---|---|---|---|---|---|---|---|---|---|---|---|---|---|
| d'Hemecourt, Zurakowski et al. 2012 [36] | Y | N | Y | N/A | N/A | N/A | N | N/A | Y | Y | Y | N | Y | 66.6 |
| Naghdi, Nakhostin Ansari et al. 2015 [37] | Y | N | Y | N/A | N/A | N/A | Y | N/A | Y | Y | N | N | N | 55.5 |
| Naghdi, Ansari et al. 2016 [67] | Y | N | Y | N/A | N/A | N/A | N | N/A | Y | Y | N | N | N | 44.4 |
| Noormohammadpour, Khezri et al. 2018 [22] | Y | N | N | N/A | N/A | N/A | Y | N/A | Y | Y | Y | N | Y | 66.6 |
| Naghdi, Ansari et al. 2015 [68] | Y | N | Y | N/A | N/A | N/A | N | N/A | N | Y | N | N | Y | 44.4 |

N/A: Not applicable or not assessed; 1. sample description; 2. sample characteristics; 3. explanation of the reference standard; 4. inter-rater blindness; 5. intra-rater blindness 6. rater or participant randomization; 7. data collection period; 8. period between repeated measures; 9. the study test is not included in the gold standard; 10. description of the data collection procedures of the experimental test; 11. description of the data collection procedures of the gold standard; 12. description of cases of sample loss; 13. suitable statistical method. Y: yes, N: no

characteristics of the participants, item 3 assesses the suitability of the study design, item 7 assesses the data collection period, items 9 to 11 examine the details of the procedure and, lastly, item 13 evaluates the statistical details.

The authors of the tool used to assess the methodological quality information such that if insufficient information was provided in some items, the item would be marked with "no [45]." Details on the articles are outlined in Table 3.

The MFS was validated as a back-specific instrument for the functional assessment of young athletes in 2012 by a research group of the Division of Sports Medicine of the Children's Hospital of Boston. This study included 94 athletes: 44 and 50 with and without low-BP, respectively. This instrument has 5 items and gathers information on three domains: symptoms (degree to which BP affects sports activity), activities of daily living (degree to which pain is associated with back extension and/or upright activities, sitting and/or flexion activities, and jumping) and lastly, a VAS for pain assessment. The authors stated certain limitations in the validation of the instrument, such as unequal distribution by age (between individuals with BP and controls in the younger age group of athletes) and by sex and the absence of correlation of MFS with minimum, moderate, and severe scores. At the end of the study, MFS was considered a valid instrument to assess the pain and functional levels of young athletes [36].

The reliability and validity of the PFRI for athletes with low-BP were examined in 2015 by a research group of the Tehran University of Medical Sciences. In total, 100 athletes with low-BP and 50 healthy athletes participated in the study. This instrument contains 10 items and measures the pain and function from 0 (no pain or can do all activities) to 4 (the worst possible pain/ or cannot do any activity) [111]. The authors indicated a limitation in the validation of the instrument: the responsiveness of PFRI was not examined. Nevertheless, they concluded that PFRI is a valid and reliable instrument for assessing the functional status of athletes with low BP [37].

A year later, the same research group validated the cross-culturally adapted PFRI for assessing athletes with neck pain [67]. In total, 100 athletes with neck pain and 50 healthy athletes participated in this study. Among the study limitations, the authors reported that the effect size-based responsiveness of the PFRI to detect changes over time was not evaluated and that they evaluated only the Persian version of the instrument. Notwithstanding these limitations, at the end of the study, the PFRI was considered a valid and reliable instrument for assessing the functional status of athletes with neck pain [67].

The ADI was recently analyzed by a group of researchers from two Iranian Universities and Stanford University [22], who assessed the validity and reliability of ADI. In total, 165 male and female athletes participated in this study. ADI contains 12 questions covering pain

intensity, stretching and strengthening or weight training exercises, sport-specific moves or skills, and movement involving back rotations, among other questions. As a study limitation, the authors reported that the intraclass correlation coefficient of the question concerning sexual activity could not be calculated. However, they were able to identify a correlation between ADI and MFS. At the end of the study, the authors concluded that ADI is a reliable and valid instrument for assessing disability in athletes with BP [22].

## 4 Discussion

Some systematic reviews have previously investigated the existence of instruments for evaluating BP [112–114]. Systematic reviews organize data objectively and may report significant results that potentially contribute to various studies. Therefore, reviews that systematize instruments for assessing pain in specific populations are relevant because these results may aid researchers in better selection of instruments according to their research objective. The main purpose of this study was to identify the instruments used in the literature to assess BP in athletes and summarize the articles that developed questionnaires for athletes. The hypothesis was that few instruments are valid to assess BP in athletes and that the available instruments contribute little to the specific assessment of BP in this population.

This systematic review summarized the results of research on BP assessment instruments in 12,912 athletes from 24 countries that were part of the studies. The studies included in this review used different instruments to assess the pain for athletes in different sports, training levels, ages, sexes, and ethnicities. The main results demonstrated that 24 (43.7%) articles used ODI and 14 (25.4%) articles used SNQ; i.e., 69% of all the articles used non-validated instruments. These two instruments were also found in other systematic reviews [13, 23, 114]. Furthermore, many instruments for assessing BP were not specifically developed to assess athletes and may have been used in several studies owing to the limited number of instruments available for this population, according to the results presented above.

In addition, 6 (46.15%), 5 (38.4%), 1 (7.7%), and 1 (7.7%) instrument(s) identified in this review were developed in Europe, North America, South America, and Asia, respectively. These results demonstrated the increased interest of developed countries in this topic and instruments developed for BP research. Considering the study site of the articles, 40 (72.7%) studies were conducted in Europe. These results reveal that countries from other continents, such as the American and African countries, require further research and studies for developing instruments for athletes.

For methodological quality assessment remained 5 articles. Another review on questionnaires [115] also identified a low number of articles for methodological quality assessment. In this research, these five articles validated the following questionnaires: MFS [36, 68], PFRI [37, 67], and ADI [22]. Upon evaluating each instrument individually, it was realized that the instruments differed substantially in terms of items and extent to which their psychometric properties were evaluated; thus, each instrument possesses an objective and functionality.

The MFS is considered a valid instrument for assessing the pain and functional levels in young athletes. Conversely, the PFRI is considered a valid and reliable tool for assessing the functional status in athletes with low-back and neck pain. Lastly, the ADI is a reliable and valid instrument for assessing disability in athletes with BP.

In this review, only two of these three (66.6%) studies selected for methodological quality assessment were classified as having high quality [22, 36] compared to other studies, in which no article was rated as a high-quality study [116, 117]. Moreover, the checklist proposed by Brink & Louw evaluates the quality of the articles and not their measuring instruments.

Therefore, the lack of evidence for BP assessment instruments, in general, is not evaluated in this research.

Limitations and strengths should be highlighted. First, the instruments used for assessing BP in athletes lack standardization and are thus heterogeneous. The lack of standardization leads researchers and professionals to use less accurate evaluation strategies. Moreover, the lack of standardization makes it difficult to identify faults and problems, can lead to recurring errors, generates excessive work and rework, and can have low reliability. Second, because the instruments assess different factors, such as pain intensity, functional capacity, physical disability, quality of life, musculoskeletal disorders, pain monitoring, etc., their selection may be complex and hinder certain studies. Third, this review did not assess the quality of the instruments. To find the right tool for a survey, it is necessary, for example, to decide what to measure and then select the most suitable instrument for measuring the results. The use of tools to improve the way research is carried out has a direct impact on the reliability of the results.

Moreover, in this search strategy, it was not including the names or acronyms of specific pain questionnaires. Thus, although some articles might have used questionnaires, such as ODI, VAS, and SNQ, concomitantly with other questionnaires, it was chosen not to specify them in the searches to broaden the results. Therefore, this choice was considered a limitation of the search strategy.

Considered as strengths of this review the fact that this article is the first that summarizes instruments that assess BP specifically for athletes. Furthermore, this review selected articles in all languages and without fixing a publication date, and these criteria are relevant because they do not limit the ability to identify articles in the literature. Lastly, the search strategy was conducted in five recognized databases (Embase, MEDLINE, SPORTDiscus, CINAHL, and Scopus).

Another strength of this review is that it presents a protocol previously structured by the same group of researchers, which synthesizes the authors' ideas for carrying out a systematic review. Moreover, this review reduces controversies in the literature, since it does not result from the number of studies in favor of a particular intervention, but form the sum of several case studies, identified and published in specific databases that are widely used in the literature. Finally, another important positive element to be highlighted is the fact that this review directs future studies to different areas of activity, it can be used as a tool to contribute to research in medicine, physiotherapy, sport, among others.

Future studies should develop instruments or a standard questionnaire [13, 118] assessing the factors associated with back and neck pain and even analyze the changes in body posture focusing on activities of daily living. Questionnaires examining these factors may promote a more robust and detailed data collection. Here are some recommendations for creating a new instrument for assessing BP in athletes in the following text.

Initially, researchers should establish the conceptual framework of what they intend to measure, in addition to defining the objectives of the questionnaire and the study population by thoroughly reading books, articles, and other materials [31] related to BP and athletes. The items should subsequently be derived from a central objective [31] and associated with the sporting activities of the athletes. These items may gather information related to several factors, such as the sport, training or competition volume, trunk rotation or flexion movements, repetitive movements, other efforts, skill-related fitness components, and rest. It is also essential to list information on behavioral care, posture and back or neck pain. These items can be either selected from items of previous instruments and adapted or created.

Subsequently, the individual content of each item should be evaluated by content validation, thereby assessing the text and improving the structure of the questionnaire. At the end, a pre-test should be performed before assessing the reliability of the questionnaire. Validity is

the extent to which an instrument measures the construct it intends to measure [119], and reliability is related to the coherence and consistency of the results and to the confidence that the test provides when measuring phenomena [120, 121]. Therefore, these measurement properties should be assessed when designing questionnaires. Finally, based on the results identified in this review, if future research wants to investigate BP in athletes, it is suggested the Athlete Disability Index because it contains 12 general questions about pain, and its results are broader compared to other instruments such as the Micheli Functional Scale.

## 5 Conclusion

This review identified specific questionnaires for assessing BP and studies that validated specific questionnaires for athletes. Based on this systematic review, two questionnaire validation articles were classified with a high methodological quality. These findings highlight the importance of a more careful evaluation in selecting a questionnaire and, once selected, in avoiding methodological shortcomings. Therefore, researchers should select the instrument according to their research objective and comply with processes, such as validity and reliability.

Even though the results demonstrate the lack of certain essential elements for good evaluation of research methodology, the importance of each instrument and their high correlation with other instruments considered "gold standards" should be emphasized. To future readers and researchers in the area, it is suggested new research articles and instruments, because the effectiveness of treatment and its follow-up depend on a reliable and valid pain assessment and measurement. It is suggested that more precise and reliable tools for assessing pain in this population be developed in the future and that these tools can be used to make the assessment of back pain more precise and standardized.

## Supporting information

**S1 Annex. Search strategies used by the respective database.**
(DOCX)

**S1 Checklist. PRISMA 2020 checklist.**
(DOCX)

## Acknowledgments

The authors would like to show their gratitude to Instituto Federal Goiano, Universidade Federal de Goiás and the Research Group on Child and Adolescent Health (www.gpsaca.com.br (accessed on 04 October 2023)) for the support.

## Author Contributions

**Conceptualization:** Vinicius Diniz Azevedo, Regina Márcia Ferreira Silva, Silvia Cristina de Carvalho Borges, Michele da Silva Valadão Fernades, Vicente Miñana-Signes, Manuel Monfort-Pañego, Priscilla Rayanne E. Silva Noll, Matias Noll.

**Data curation:** Vinicius Diniz Azevedo, Regina Márcia Ferreira Silva, Silvia Cristina de Carvalho Borges, Michele da Silva Valadão Fernades, Vicente Miñana-Signes, Manuel Monfort-Pañego, Priscilla Rayanne E. Silva Noll, Matias Noll.

**Formal analysis:** Vinicius Diniz Azevedo, Regina Márcia Ferreira Silva, Silvia Cristina de Carvalho Borges, Michele da Silva Valadão Fernades, Vicente Miñana-Signes, Manuel Monfort-Pañego, Priscilla Rayanne E. Silva Noll, Matias Noll.

**Funding acquisition:** Vinicius Diniz Azevedo, Regina Márcia Ferreira Silva, Silvia Cristina de Carvalho Borges, Michele da Silva Valadão Fernades, Vicente Miñana-Signes, Manuel Monfort-Pañego, Priscilla Rayanne E. Silva Noll, Matias Noll.

**Investigation:** Vinicius Diniz Azevedo, Regina Márcia Ferreira Silva, Silvia Cristina de Carvalho Borges, Michele da Silva Valadão Fernades, Vicente Miñana-Signes, Manuel Monfort-Pañego, Priscilla Rayanne E. Silva Noll, Matias Noll.

**Methodology:** Vinicius Diniz Azevedo, Regina Márcia Ferreira Silva, Silvia Cristina de Carvalho Borges, Michele da Silva Valadão Fernades, Vicente Miñana-Signes, Manuel Monfort-Pañego, Priscilla Rayanne E. Silva Noll, Matias Noll.

**Project administration:** Vinicius Diniz Azevedo, Regina Márcia Ferreira Silva, Silvia Cristina de Carvalho Borges, Michele da Silva Valadão Fernades, Vicente Miñana-Signes, Manuel Monfort-Pañego, Priscilla Rayanne E. Silva Noll, Matias Noll.

**Resources:** Vinicius Diniz Azevedo, Regina Márcia Ferreira Silva, Silvia Cristina de Carvalho Borges, Michele da Silva Valadão Fernades, Vicente Miñana-Signes, Manuel Monfort-Pañego, Priscilla Rayanne E. Silva Noll, Matias Noll.

**Software:** Vinicius Diniz Azevedo, Regina Márcia Ferreira Silva, Silvia Cristina de Carvalho Borges, Michele da Silva Valadão Fernades, Vicente Miñana-Signes, Manuel Monfort-Pañego, Priscilla Rayanne E. Silva Noll, Matias Noll.

**Supervision:** Vinicius Diniz Azevedo, Regina Márcia Ferreira Silva, Silvia Cristina de Carvalho Borges, Michele da Silva Valadão Fernades, Vicente Miñana-Signes, Manuel Monfort-Pañego, Priscilla Rayanne E. Silva Noll, Matias Noll.

**Validation:** Vinicius Diniz Azevedo, Regina Márcia Ferreira Silva, Silvia Cristina de Carvalho Borges, Michele da Silva Valadão Fernades, Vicente Miñana-Signes, Manuel Monfort-Pañego, Priscilla Rayanne E. Silva Noll, Matias Noll.

**Visualization:** Vinicius Diniz Azevedo, Regina Márcia Ferreira Silva, Silvia Cristina de Carvalho Borges, Michele da Silva Valadão Fernades, Vicente Miñana-Signes, Manuel Monfort-Pañego, Priscilla Rayanne E. Silva Noll, Matias Noll.

**Writing – original draft:** Vinicius Diniz Azevedo, Regina Márcia Ferreira Silva, Silvia Cristina de Carvalho Borges, Michele da Silva Valadão Fernades, Vicente Miñana-Signes, Manuel Monfort-Pañego, Priscilla Rayanne E. Silva Noll, Matias Noll.

**Writing – review & editing:** Vinicius Diniz Azevedo, Regina Márcia Ferreira Silva, Silvia Cristina de Carvalho Borges, Michele da Silva Valadão Fernades, Vicente Miñana-Signes, Manuel Monfort-Pañego, Priscilla Rayanne E. Silva Noll, Matias Noll.

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
