## [Decision Letter · Decision Letter 0]

9 May 2023

PONE-D-23-08713Instruments for assessing back pain in athletes: a systematic reviewPLOS ONE

Dear Dr. Ferreira Silva,

Thank you for submitting your manuscript to PLOS ONE. After careful consideration, we feel that it has merit but does not fully meet PLOS ONE’s publication criteria as it currently stands. Therefore, we invite you to submit a revised version of the manuscript that addresses the points raised during the review process.

We look forward to receiving your revised manuscript.

Kind regards,

Esedullah Akaras

Academic Editor

PLOS ONE

Journal Requirements:

3. We note that Figure 3in your submission contain [map/satellite] images which may be copyrighted. All PLOS content is published under the Creative Commons Attribution License (CC BY 4.0), which means that the manuscript, images, and Supporting Information files will be freely available online, and any third party is permitted to access, download, copy, distribute, and use these materials in any way, even commercially, with proper attribution. For these reasons, we cannot publish previously copyrighted maps or satellite images created using proprietary data, such as Google software (Google Maps, Street View, and Earth). For more information, see our copyright guidelines: http://journals.plos.org/plosone/s/licenses-and-copyright.

a. You may seek permission from the original copyright holder of Figure 3 to publish the content specifically under the CC BY 4.0 license.  

Additional Editor Comments:

Dear Regina Márcia Ferreira Silva,

Major Revisions have been requested.

Reviewers' comments:

Reviewer's Responses to Questions

**Comments to the Author**

1. Is the manuscript technically sound, and do the data support the conclusions?

Reviewer #1: Yes

Reviewer #2: Partly

2. Has the statistical analysis been performed appropriately and rigorously? 

Reviewer #1: Yes

Reviewer #2: N/A

3. Have the authors made all data underlying the findings in their manuscript fully available?

Reviewer #1: Yes

Reviewer #2: Yes

4. Is the manuscript presented in an intelligible fashion and written in standard English?

Reviewer #1: Yes

Reviewer #2: Yes

5. Review Comments to the Author

Reviewer #1: Dear authors,

I am glad to have had the opportunity to review this study. Congratulations, it's a nice study with clinical outputs. I think that your research will contribute to the literature. I hope the comments contribute to your study. It will look better with some corrections. Study needs corrections and the authors should carefully address the following issues:

- The article type was entered incorrectly, please correct it.

- Pay attention to punctuation marks and spaces between characters.

- Replace informal uses (we, our, etc.).

- Annex or appendix? Prefer single use.

- There is no appendix 2.

Abstract

- Generally acceptable.

Introduction

- Generally acceptable.

Materials and Methods

- 2.6 Reviewer training: “Lastly, the training process…” Detail or reference this process.

- Spelling errors can be seen throughout the text. I will mention some of it, but it should be reviewed on this.

Results

- Page 12, line 180: Pay attention to dots and parentheses.

- In Table 2, the reliability of reference 37 is given as Cronbach's alpha, check it.

- Page 24: Delete the extra comma in the sentence above Table 3.

Discussion:

- Generally acceptable.

- Also in the last paragraph, can you tell which instruments you would recommend using for pain or disability based on these results?

Reviewer #2: Overall, I would like to appreciate the authors for their effort to provide this important and interesting work.

This manuscript titled "Instruments for assessing back pain in athletes: a systematic review". Accordingly, the authors should focus on the study question "Back pain in athletes". However, the authors started abstract with back and neck pain.

Introduction: The authors should explain the importance of the study in detail. The authors should add the hypothesis of the study.

Discussion: The strengths of this review should be demonstrated with detailed explanation. Future directions and recommendations need to be clarified.

Conclusion: Add a brief message to readers and researchers.

6. PLOS authors have the option to publish the peer review history of their article (what does this mean?). If published, this will include your full peer review and any attached files.

Reviewer #1: **Yes: **Halime ARIKAN

Reviewer #2: No

---

## [Author Response · Author response to Decision Letter 0]

20 Jul 2023

Reviewer #1

We would like to express our thanks to the editor and two reviewers for their thoughtful and in-depth comments concerning our manuscript. Your suggestions helped us improve the quality of our paper. We carefully considered every comment and made all the appropriate changes. Reviewer comments are detailed first, and our responses to the points raised follow (italics). We have highlighted the changes made to the body of the manuscript using red coloured text in this document.

Additional Editor Comments

 1. I am glad to have had the opportunity to review this study. Congratulations, it's a nice study with clinical outputs. I think that your research will contribute to the literature. I hope the comments contribute to your study. It will look better with some corrections. Study needs corrections and the authors should carefully address the following issues

Author´s response: Thank you very much for your positive feedback. 

 2. The article type was entered incorrectly, please correct it.

Author´s response: Thanks for your careful review. We made the correction as requested at the time of resubmission of the manuscript with corrections. 

 3. Pay attention to punctuation marks and spaces between characters.

Author´s response: Thanks for your in-depth review.

 4. Replace informal uses (we, our, etc.).

Author´s response: Thanks for your careful review. Informal usage has been replaced throughout the text, below are some examples:

“Therefore, the aim of this study was to review the scientific literature related to validated questionnaires to evaluate BP in athletes.” (Lines 81-83)

“In this review, only two of these three (66.6%) studies selected for methodological quality assessment were classified as having high quality (36, 37) compared to other studies, in which no article was rated as a high-quality study (120, 121). (Lines 374-376)

“Limitations and strengths should be highlighted.” (Line 380)

 5. Annex or appendix? Prefer single use.

Author´s response: Thanks for your in-depth review. The title was corrected.

“Annex 1” (Page 31)

 6. There is no appendix 2.

Author´s response: Thanks for your careful review. We agree with the reviewer 

The mention to “appendix 2” was removed, and replaced by the phrase " The search strategy for each database is presented in Annex 1" (Line 111).

Abstract

 7. Generally acceptable.

Author´s response: Thank you very much for your positive feedback.

Introduction

 8. Generally acceptable.

Author´s response: Thanks.

Materials and Methods

 9. 2.6 Reviewer training: “Lastly, the training process…” Detail or reference this process.

Author´s response: Thanks for your careful review. A brief excerpt of the process details is included as follows:

“The reviewers were trained on how to use the software to use the same selection criteria for the articles. In this case, at first, a reviewer included in the software a file with data from the articles and began the selection of those that fit the selection criteria by reading the titles and abstracts, including justifying the exclusions. Later, the second reviewer performed the same procedure.” (Lines 161-165)

 10. Spelling errors can be seen throughout the text. I will mention some of it, but it should be reviewed on this.

Author´s response: Thanks for your in-depth review. We performed a complete spelling correction.

Results

 11. Page 12, line 180: Pay attention to dots and parentheses.

Author´s response: Thanks for your careful review. The error has been corrected, as follows:

“The terms that defined the athletes’ competitive level, such as educational (education, university student, elite sports school), divisional (international, national, regional, and local), age (veterans), competition (professional or elite and recreational (club and intramural sports and Junior Olympic program) levels varied considerably between the studies.” (Lines 189-193)

 12. In Table 2, the reliability of reference 37 is given as Cronbach's alpha, check it.

Author´s response: Thanks for your in-depth review. We made de correction..

 13. Page 24: Delete the extra comma in the sentence above Table 3.

Author´s response: Thanks for your careful review. The error has been corrected, as follows:

“The authors of the tool used to assess the methodological quality information such that if insufficient information was provided in some items, the item would be marked with “no (46).” Details on the articles are outlined in Table 3.” (Lines 296-298)

Discussion:

 14. Generally acceptable.

Author´s response: Thanks a lot. 

 15. Also in the last paragraph, can you tell which instruments you would recommend using for pain or disability based on these results?

Author´s response: Thanks for your careful review. Follow our recommendation:

“Finally, based on the results identified in this review, if future research wants to investigate BP in athletes, it is suggested the Athlete Disability Index because it contains 12 general questions about pain, and its results are broader compared to other instruments such as the Micheli Functional Scale.” (Lines 415-418)

 16. Overall, I would like to appreciate the authors for their effort to provide this important and interesting work.

Author´s response: Thanks you very much. We are happy to know that our work is being recognized. 

Reviewer #2

We would like to express our thanks to the editor and two reviewers for their thoughtful and in-depth comments concerning our manuscript. Your suggestions helped us improve the quality of our paper. We carefully considered every comment and made all the appropriate changes. Reviewer comments are detailed first, and our responses to the points raised follow (italics). We have highlighted the changes made to the body of the manuscript using red coloured text in this document.

Additional Editor Comments

 1. Overall, I would like to appreciate the authors for their effort to provide this important and interesting work

Author´s response: Thank you very much for your positive feedback. 

 2. This manuscript titled "Instruments for assessing back pain in athletes: a systematic review". Accordingly, the authors should focus on the study question "Back pain in athletes". However, the authors started abstract with back and neck pain.

Author´s response: Thanks for your careful review. We agree with the reviewer. We removed the word as suggested, as follow:

“Back pain in athletes varies with sport, age, and sex, which can impair athletic performance, thereby contributing to retirement.” (Lines 15-16)

Introduction

 3. The authors should explain the importance of the study in detail. 

Author´s response: Thanks for your in-depth review. The importance of this study was highlighted in more details, as follow:

“This study is relevant because the demand in health care, combined with the need to use evaluative instruments, has contributed to increase, for example, the pressure on health care professionals to ensure the implementation of evidence-based practice. Thus, publication of systematic review studies, as well as others that synthesize research results, is an important action for evidence-based practice. Moreover, this type of study serves to guide the development of projects, indicating new directions for future investigations.” (Lines 76-81)

 4. The authors should add the hypothesis of the study.

Author´s response: Thanks for your careful review. We also include our hypothesis in the introduction, as follow:

“Therefore, the aim of this study was to review the scientific literature related to validated questionnaires to evaluate BP in athletes. The hypothesis is that there are few valid and reliable instruments to evaluate BP in athletes and that the existing ones contribute little to the specific assessment.” (Lines 81-84).

Discussion

 5. The strengths of this review should be demonstrated with detailed explanation. 

Author's Response: Thanks for your in-depth review. The strengths of this review are demonstrated, as follows:

“Considered as strengths of this review the fact that this article is the first that summarizes instruments that assess BP specifically for athletes. Furthermore, this review selected articles in all languages and without fixing a publication date, and these criteria are relevant because they do not limit the ability to identify articles in the literature. Lastly, the search strategy was conducted in five recognized databases (Embase, MEDLINE, SPORTDiscus, CINAHL, and Scopus).” (Lines 389-394).

Conclusion

 6. Future directions and recommendations need to be clarified.

Author´s response: Thanks for your careful review. Message to readers and researchers was reinforced, as follows:

“To future readers and researchers in the area, we suggest new research articles and instruments, because the effectiveness of treatment and its follow-up depend on a reliable and valid pain assessment and measurement.” (Lines 431-434)

---

## [Decision Letter · Decision Letter 1]

17 Aug 2023

PONE-D-23-08713R1Instruments for assessing back pain in athletes: a systematic reviewPLOS ONE

Dear Dr. Ferreira Silva,

Thank you for submitting your manuscript to PLOS ONE. After careful consideration, we feel that it has merit but does not fully meet PLOS ONE’s publication criteria as it currently stands. Therefore, we invite you to submit a revised version of the manuscript that addresses the points raised during the review process.

We look forward to receiving your revised manuscript.

Kind regards,

Esedullah Akaras

Academic Editor

PLOS ONE

Journal Requirements:

Additional Editor Comments:

Your publication will be accepted after making the relevant minor corrections.

Reviewers' comments:

Reviewer's Responses to Questions

**Comments to the Author**

1. If the authors have adequately addressed your comments raised in a previous round of review and you feel that this manuscript is now acceptable for publication, you may indicate that here to bypass the “Comments to the Author” section, enter your conflict of interest statement in the “Confidential to Editor” section, and submit your "Accept" recommendation.

Reviewer #1: All comments have been addressed

2. Is the manuscript technically sound, and do the data support the conclusions?

Reviewer #1: Yes

3. Has the statistical analysis been performed appropriately and rigorously? 

Reviewer #1: Yes

4. Have the authors made all data underlying the findings in their manuscript fully available?

Reviewer #1: Yes

5. Is the manuscript presented in an intelligible fashion and written in standard English?

Reviewer #1: Yes

6. Review Comments to the Author

Reviewer #1: The requested corrections have been corrected carefully but can be published with a correction of a few minor points. So thank you and congratulations to the authors.

1. Informal uses still exist in the text. Please correct using the "Find" option one by one.

2. The Reference 37's reliability stands as Cronbach alpha. Internal consistency is also an indicator of reliability, but this is included in the previous column. If ICC is not available, indicate it.

7. PLOS authors have the option to publish the peer review history of their article (what does this mean?). If published, this will include your full peer review and any attached files.

Reviewer #1: No

---

## [Author Response · Author response to Decision Letter 1]

1 Sep 2023

Reviewer #1: 

The requested corrections have been corrected carefully but can be published with a correction of a few minor points. So thank you and congratulations to the authors.

Author´s response: Thanks for your careful review.

 1. Informal uses still exist in the text. Please correct using the "Find" option one by one.

Author´s response: Thanks for your careful review. Informal usage has been substituted in several passages of the text, as follow:

“To support this review, was carried out and published an article on a systematic review…” (Lines 89-90)

“Moreover, research has been carried out in other countries, such as in Norway…”(Lines 227-228)

“This research identified 24 (43.7%) articles using ODI (103) and 14 (25.4%) using SNQ (104).” (line 258)

“Here are some recommendations for creating a new instrument for assessing BP in athletes in the following text.” (line 399)

“…, it is suggested new research articles and instruments, …” (line 433)

 2. The Reference 37's reliability stands as Cronbach alpha. Internal consistency is also an indicator of reliability, but this is included in the previous column. If ICC is not available, indicate it.

Author´s response: Thanks for your careful review. Can in use the intraclass correlation coefficient (ICC) for test/retest reliability. However, the authors of the text did not use ICC. As reliability can be verified by Cronbach's alpha, we agree with your suggestion to indicate internal consistency as an indicator of reliability. We have therefore made the correction in Table 2.

---

## [Decision Letter · Decision Letter 2]

18 Sep 2023

PONE-D-23-08713R2Instruments for assessing back pain in athletes: a systematic reviewPLOS ONE

Dear Dr. Ferreira Silva,

Thank you for submitting your manuscript to PLOS ONE. After careful consideration, we feel that it has merit but does not fully meet PLOS ONE’s publication criteria as it currently stands. Therefore, we invite you to submit a revised version of the manuscript that addresses the points raised during the review process.

We look forward to receiving your revised manuscript.

Kind regards,

Esedullah Akaras

Academic Editor

PLOS ONE

Journal Requirements:

Reviewers' comments:

Reviewer's Responses to Questions

**Comments to the Author**

1. If the authors have adequately addressed your comments raised in a previous round of review and you feel that this manuscript is now acceptable for publication, you may indicate that here to bypass the “Comments to the Author” section, enter your conflict of interest statement in the “Confidential to Editor” section, and submit your "Accept" recommendation.

Reviewer #1: All comments have been addressed

Reviewer #2: All comments have been addressed

2. Is the manuscript technically sound, and do the data support the conclusions?

Reviewer #1: Yes

Reviewer #2: Yes

3. Has the statistical analysis been performed appropriately and rigorously? 

Reviewer #1: Yes

Reviewer #2: Yes

4. Have the authors made all data underlying the findings in their manuscript fully available?

Reviewer #1: Yes

Reviewer #2: Yes

5. Is the manuscript presented in an intelligible fashion and written in standard English?

Reviewer #1: Yes

Reviewer #2: Yes

6. Review Comments to the Author

Reviewer #1: The authors have made appropriate corrections. I am sure it will contribute to the literature and the journal.

Reviewer #2: I would like to thank the authors for their responses to all previous comments. However, minor comments still need to be addressed as follows:

- Discuss the implications and strengths of this review. Explain the limitations in detail.

- Conclusion: add a brief message ".............." to readers and researchers.

7. PLOS authors have the option to publish the peer review history of their article (what does this mean?). If published, this will include your full peer review and any attached files.

Reviewer #1: No

Reviewer #2: No

---

## [Author Response · Author response to Decision Letter 2]

5 Oct 2023

PLOS ONE Decision: Revision required [PONE-D-23-08713R2] - [EMID:96f974367d29e64a]

Instruments for assessing back pain in athletes: a systematic review

PLOS ONE

Dear Dr. Ferreira Silva,

Thank you for submitting your manuscript to PLOS ONE. After careful consideration, we feel that it has merit but does not fully meet PLOS ONE’s publication criteria as it currently stands. Therefore, we invite you to submit a revised version of the manuscript that addresses the points raised during the review process.

We would like to express our thanks to the editor and two reviewers for their thoughtful and in-depth comments concerning our manuscript. Your suggestions helped us improve the quality of our paper. We carefully considered every comment and made all the appropriate changes. Reviewer comments are detailed first, and our responses to the points raised follow (italics). We have highlighted the changes made to the body of the manuscript using red coloured text.

Additional Editor Comments

Author´s response: The reference list was completely checked. We also inform you that no retracted reference was used in this manuscript. We also inform you that the manuscript was revised in English by native speaker.

Reviewer #2: 

 • Would like to thank the authors for their responses to all previous comments.

Author´s response: Thanks for your careful review

 • Discuss the implications and strengths of this review. 

Author´s response: Thanks for your careful review. A paragraph has been included which highlights more of the strengths of this review, as follow:

Another strength of this review is that it presents a protocol previously structured by the same group of researchers, which synthesizes the authors’ ideas for carrying out a systematic review. Moreover, this review reduces controversies in the literature, since it does not result from the number of studies in favor of a particular intervention, but form the sum of several case studies, identified and published in specific databases that are widely used in the literature. Finally, another important positive element to be highlighted is the fact that this review directs future studies to different areas of activity, it can be used as a tool to contribute to research in medicine, physiotherapy, sport, among others. (Lines 403-410)

 • Explain the limitations in detail.

Author´s response: Thanks for your careful review. Sentences that further highlight the limitations of this review have been included, as follows:

The lack of standardization leads researchers and professionals to use less accurate evaluation strategies. Moreover, the lack of standardization makes it difficult to identify faults and problems, can lead to recurring errors, generates excessive work and rework, and can have low reliability. (Lines 381-384)

To find the right tool for a survey, it is necessary, for example, to decide what to measure and then select the most suitable instrument for measuring the results. The use of tools to improve the way research is carried out has a direct impact on the reliability of the results. (Lines 388- 391)

 • Conclusion: add a brief message “………” to readers and researchers

 Author´s response: Thanks for your careful review. Periods that further highlight the limitations of this review have been included, as follows:

It is suggested that more precise and reliable tools for assessing pain in this population be developed in the future and that these tools can be used to make the assessment of back pain more precise and standardized. (Lines 450-452)

---

## [Editor Report · Decision Letter 3]

11 Oct 2023

Instruments for assessing back pain in athletes: a systematic review

PONE-D-23-08713R3

Dear Dr. Ferreira Silva,

We’re pleased to inform you that your manuscript has been judged scientifically suitable for publication and will be formally accepted for publication once it meets all outstanding technical requirements.

Kind regards,

Esedullah Akaras

Academic Editor

PLOS ONE
---

## [Editor Report · Acceptance letter]

25 Oct 2023

PONE-D-23-08713R3 

Instruments for assessing back pain in athletes: a systematic review 

Dear Dr. Ferreira Silva:

I'm pleased to inform you that your manuscript has been deemed suitable for publication in PLOS ONE. Congratulations! Your manuscript is now with our production department. 

Kind regards, 

on behalf of

Dr. Esedullah Akaras 

Academic Editor

PLOS ONE